# Valorization of a Waste Product of Edible Flowers: Volatile Characterization of Leaves

**DOI:** 10.3390/molecules27072172

**Published:** 2022-03-27

**Authors:** Basma Najar, Laura Pistelli, Ilaria Marchioni, Luisa Pistelli

**Affiliations:** 1Dipartimento Scienze Agrarie, Alimentari e Agro-ambientali (DISAAA-a), Università di Pisa, 56124 Pisa, Italy; laura.pistelli@unipi.it (L.P.); ilaria.marchioni.16@gmail.com (I.M.); 2Centro Interdipartimentale di Ricerca Nutraceutica e Alimentazione per la Salute (NUTRA-FOOD), Università di Pisa, 56124 Pisa, Italy; luisa.pistelli@unipi.it; 3Dipartimento di Farmacia, Università di Pisa, 56126 Pisa, Italy

**Keywords:** Lamiaceae, Mentheae tribe, Ocimeae tribe, Salviinae subtribe, Nepetinae subtribe, Menthinae subtribe, volatile organic compounds, essential oils, statistical analyses

## Abstract

(1) Background: The leaves of some plants are reported for their culinary uses, while in edible flowers, they are one of the discarded products in the supply chain. We investigated the volatile profile (VP) and the essential oil (EO) compositions of leaves from 12 Lamiaceae species, of which nine belong to the Mentheae tribe and three to the Ocimeae tribe. (2) Methods: Phytochemical analyses were performed using a GC-MS instrument. (3) Results: More than 53% of the Ocimeae tribe VP was represented by sesquiterpene hydrocarbons (SH), followed by phenylpropanoids, except for *O.* × *citriodorum,* where oxygenated monoterpenes (OM) were the second main class. OM prevailed in six species of the Mentheae tribe except for *Agastache* ‘Arcado Pink’, *Salvia discolor*, and *S. microphylla,* where SH dominated. The EO composition of Ocimeae tribe showed a similar behavior to that of VP concerning the predominant classes. *O. basilicum* ‘Blue Spice’ (Ob-BS) was an exception, since it showed oxygenated sesquiterpenes (OS: 29.6%) as a second principal class. Sesquiterpene compounds were also present in a high amount in two species of the Salviinae subtribe (*S. microphylla* and *S. discolor*) and two of the Nepetinae subtribe (*Nepeta* × *faasenii* and *A.* ‘Arcado Pink’). The remaining species of the Mentheae tribe were characterized by OM. (4) Conclusions: Many of the main compounds found were reported for their importance in human health and thus are important as ingredients in several new industrial products.

## 1. Introduction

Since the time of the Neanderthals, humans used leaves as food and medicine. In addition to the use of leaves from herbs in folk medicine, cosmetics, and other industrial uses, edible leaves represent an important element in the daily diet of humans and livestock [1]. From a culinary point of view, leaves have also been used rudimentarily to cook and package food, adding flavor and color. Cooking with leaves gained more popularity not only for gastronomic intake but also for merging health benefits with the flavors of a variety of leaves. According to Motti [1], leaves are the plant organs most frequently used as vegetables in Italy and contain high concentrations of bioactive metabolites [2].

In recent decades, and as a result of the search for new natural food products, the consumption of new edible plant species and organs (e.g., edible flowers, EFs) has been rediscovered, and has become a trend in the food industry, with an increase in the production of EFs. Despite this rise in consumption, it is still considered part of a niche market and suffers from some limitations (seasonality and perishability of the flowers, and high cost of production owing to the high standard of quality required) which imposed a high level of selection and so leads to a high discard amount [3], mainly of other plant organs, such as leaves [4]. The management of waste and by-products is a major concern in the world [5] and has an impact on economics and social sectors, but mainly on the environment [6].

During the ANTEA project (Interreg ALCOTRA “ANTEA” project (N° 1139)), whose objective was the study of edible flowers, it was possible to investigate the potential use of the waste products from this supply chain, mainly constituted by leaves, to make the production process sustainable and to recycle the majority of the produced biomass. Many of the selected leaves are already used as food ingredients or in infusions, but for some of them this use is not feasible, due to the morphology of the leaves, taste, and palatability. Because of this, an investigation of the volatile composition of discarded leaves is important to address this plant material for other industries, exploiting their potential as ‘secondary raw material’ able to produce essential oil useful as a flavoring product, cosmetic ingredient, or environmental disinfectant, and meanwhile avoiding wasting resources and time for the disposal of the leaves. Furthermore, the good agriculture practices and the organic methods used for the edible flower production yield to a good waste product, free from contaminants, ready to be processed without any preliminary preparation.

The Lamiaceae family (mint), a large botanical family with more than 236 genera and 7000 species [5], is considered to be one of the most important plant families in the world and the most studied [6]. Many species are cultivated for their fragrant leaves and attractive flowers [7] and are widely known also for their use in folk medicine as herbal remedies for various disorders [8,9]. From the culinary point of view, some species of mint family are used as aromatic herbs and cultivated for their edible leaves (ELs). The use of *Ocimum basilicum* leaves is varied and includes both culinary and pharmaceutical manufacture [10]. Catnip (*Nepeta* genus) young leaves are reported to be edible and have an aromatic mint-like flavor, eaten in salad [11]. The preparation of leaf infusion is especially related to several *Salvia* spp. [12] as well as the North American indigenous bee balm species (*Monarda* genus) [13]. The use of either fresh or dried *Agastache* leaves is related mostly to the *A. foeniculum* species with its pleasant scent of mint, used to flavor cakes, bread, or chicken, and for preparing refreshing tea [14].

This work investigated the aromatic profile of 12 leaves from edible flowers of Lamiaceae species, related to the subfamily Nepetoideae. Nine of them belong to Mentheae tribe of which four are from the Nepetinae subtribe (*Agastache* ‘Arcado Pink’, *Agastache aurantiaca* (A.Gray) Lint & Epling, *Agastache* ‘Blue Boa’, *Agastache mexicana* (Kunth) Lint & Epling and *Nepeta* × *faasenii* Bergmans ex Stearn); one from the Menthinae subtribe (*Monarda didyma* L.) and three from the Salviinae subtribe (*Salvia discolor* Kunth, *Salvia dorisiana* Standl, and *Salvia microphylla* Kunth). Moreover, three plants belong to the Ocimeae tribe: *Ocimum basilicum* ‘Blue Spice’; *Ocimum basilicum* ‘Cinnamon’ L., *Ocimum × citriodorum* Vis were also studied.

## 2. Results and Discussion

### 2.1. HS-SPME Analysis

#### 2.1.1. Mentheae Tribe

Seventy-six compounds were found in VOCs of the studied leaves representing at least 97% of the identified fraction (Appendix A). The *Agastache* genus was characterized by monoterpenes, especially oxygenated compounds (OM), which prevailed in *A. mexicana* (83.1%) and their amount exceeded 50% in both *A. aurantiaca* (56.5%) and *A.* ‘Blue Boa’ (57.9%). In contrast, *A.* ‘Arcado Pink’ leaves were rich in sesquiterpene hydrocarbons (92.2%). More specifically, geranyl acetate alone (N° 31) represented more than the half of the OM present in *A. mexicana*, while it was completely absent in the other two *Agastache* species and replaced by pulegone (N° 16), a compound with a strong pungent aromatic mint smell (51.9% in *A.* ‘Blue Boa’ vs. 52.8% in *A. aurantiaca*). *A.* ‘Arcado Pink’ demonstrated both germacrene D (N° 45; 33.3%) and β-caryophyllene (N° 37; 21.6%) as its main compounds. The latter was also highly present in both *A.* ‘Blue Boa’ (19.0%) and *A. aurantiaca* (35.4%). In the literature, there were few investigations on leaf VOCs for this genus, since only leaves of *A. foeniculum* (Pursh) Kuntze and *A. rugosa* (Fisch. & C.A.Mey.) Kuntze [15,16] were studied until now [15,16]. The first study reported the domination of estragol in both species. Later, the prevalence of estragol (94.35%) in *A. rugosa* was confirmed in the study by Yamani [16]. Their results disagreed with what was found herein, where this compound was completely omitted in the studied species. Wilson et al. [15] investigated the volatilome of *A.* ‘Blue Boa’ leaves and reported that this hybrid was characterized by α-limonene (80.3%), β-caryophyllene (2.0%), and δ-cadinene (3.26%). All these compounds were present in the same species studied herein, though with a lesser amount, except for β-caryophyllene, which had a greater percentage (9.8%) with respect to the literature data.

In the VOC of both *M. didyma* and *N.* × *faasenii* the chemical compounds were divided mainly between two classes: OM (56.5% and 45.0%, respectively) and SH (39.9% and 46.0% respectively). Thymol (N° 24; 37.2%) and thymol methyl ether (N° 15; 13.3%) represented most OMs in *M. didyma*, while *cis*-*trans*-nepetalactone (N° 34; 37.1%) together with eucalyptol (N° 4; 7.4%) were the main compounds in *N.* × *faasenii*. β-caryophyllene (N° 37) (9.8% in *M. didyma* vs. 34.8% in *N.* × *faasenii*) and germacrene D (N° 52) (23.0%, and 6.0%, respectively), SH constituents, were present at a considerable percentage. No reports concerning the spontaneous emission of *Monarda* genus were published until now, while few works reported the VOCs emitted by *Nepeta* genus leaves. The presence of a high amount of nepetalactone was observed in *N. sibirica* L. and especially in *N. rtanjensis* Diklic [17]. Barhoumi et al. [18] showed that the leaf emission profile of *N. curviflora* Boiss. was rich in SH especially represented by β-caryophyllene and germacrene D. This report agreed with the compounds detected in our species.

Nearly all the VOCs of *S. discolor* leaves were represented by SH (97.8%) with β-caryophyllene (53.9%) and β-elemene (N° 33) (13.4%) as the major constituents. More than 60% of the volatile fraction of *S. microphylla* was also represented by the same class previously cited and here again β-caryophyllene (27.6%) was one of the main compounds, while β-elemene was replaced by α-copaene (N° 30; 12.0%). It is also interesting to note in this species the important amount of both eucalyptol (11.1%) and camphor (N° 8; 8.4%), which constituted almost the total of oxygenated monoterpenes (OMs) (19.8%). *S. dorisiana* has a slightly different behavior since the volatile composition was divided among more chemical classes. OMs together represented 53.8%, with 32.5% SHs. Methyl perillate (N° 35; 25.2%), perillyl acetate (N° 42; 23.6%), and β-caryophyllene (23.9%) were the most important compounds. The prevalence of SH in the *Salvia* genus was also found in four out of five Iranian *Salvia* leaf species studied by Cozzolino [19], even though the main compounds mostly belonged to monoterpenes. Najar and collaborators [20] underlined the dominance of SH class on *S. broussonetii* Benth. leaves with prevalence of germacrene D and β-bourbonene.

β-caryophyllene was the unique compound shared by all the above investigated species belong to Mentheae tribe even though belonging to different subtribes. Alpha-humulene was present in eight out of nine species while limonene was also present in all the Mentheae tribe except for two *Salvia* species (*S. discolor* and *S. microphylla*). PC2 (the direction explaining the variance of 21.3%), showed a clear difference between *A. aurantiaca* and *A.* ‘Blue Boa’ and all the other species. The first two were plotted at a positive value of PC2 (right high quadrant Figure 1) and this position can be explained by the high amount of pulegone. PC1, indeed (whose direction explain the maximum variance (33.7%)), segregated *A. mexicana* from the rest and placed it in the borderline of the left high quadrant, due to its large percentage in geranyl acetate (46.7%). In this same quadrant *M. didyma* is also located; it is almost in the middle position between *A. mexicana* and *A.* ‘Arcado Pink’ due to its lowest amount of β-caryophyllene (9.8%) as *A. mexicana* (7.5%) and its highest amount of germacrene D as *A.* ‘Arcado Pink’ (23.0 and 33.3%, respectively). The left low quadrant englobes the three species of Salviinae subtribe, together with one member of Nepetinae subtribe, *N.* × *faasenii*. This position is due to their elevated percentage of β-caryophyllene which ranged from 27.6% in *S. microphylla* to exceed 53% in *S. discolor*.

The two-way HCA cluster (Figure 2) confirmed the results reported by PCA and the two clusters fit perfectly with what reported by the two axes. HCA analysis in fact was divided into two clusters: the first one ‘A’ distinguished *A. aurantiaca* and *A.* ‘Blue Boa’, as in PC2, while the second cluster ‘B’ enclosed all the remaining species. The latter can be divided into two sub-clusters where the first one, ‘1’ was homogenous and consisted of only *A. mexicana*. The second, ‘2’ subclustered Salviinae, Menthinae species together with *N.* × *faasenii*, a member of the Nepetinae sub-tribe. 

#### 2.1.2. Ocimeae Tribe

Sesquiterpene hydrocarbons (SH) represented more than 50% of the VOCs in the *Ocimeae* tribe (percentage ranged between 54.7% in *Ocimum × citriodorum* (Ob-Ct) and 56.6% in *O. basilicum* ‘Blue Spice’ (Ob-BS)). Eugenol (N° 28), a phenylpropanoid constituent, prevailed in each (Ob-Ct) and (Ob-BS) with relative percentages of 38.1% and 28.8%, respectively. In contrast, citral (neral (N° 17) + geranial (N° 21), an oxygenated monoterpene constituent, stood out (32.3%) in Ob-Ct. The high amount of geranial (N° 21), in Ob-Ct (23.7%) is noteworthy. There were many papers on the spontaneous emission of *Ocimum basilicum* (Ob) aerial parts [21,22,23,24,25,26] reported in the literature and a high presence of eugenol (41.9%) was also observed by Tirillini and Maggi [27]. Al-Kateb [28], instead, investigating lemon basil (*O. × citriodorum*), showed that the citral amount, the major contributor to the lemony flavor characteristic of lemon basil infusions, was considerably higher in the full-flowering stage (65%). This evidence was confirmed by the results of this work even though the amount was halved.

One-way PERMANOVA (Bray-Curtis similarity index) reported a statistically significant difference between both subtribe (Salviinae, Menthinae and Nepetinae sub-tribes) and tribe (Mentheae and Ocimeae tribes) (Table 1A,B) (*p*-value < 0.01 gradient).

The pairwise of these tests underlined a significant difference between Salviinae and Nepetinae subtribes from one hand and between Mentheae and Ocimeae tribes from the other hand (Table 2A1,B1).

SIMPER (SIMilarity PERcentage) analyses demonstrated that three-quarters of the subtribe differences were found in 17 compounds (Table 3A2), of which pulegone and β-caryophyllene together made up 33.5% of this dissimilarity. On the contrary, 21 compounds were characterized by an average dissimilarity higher than 1% (Table 3B2) and assumed a dissimilarity higher than 80% between tribes. Here again, β-caryophyllene and pulegone were the compounds with the most important weight in the dissimilarity between the two tribes even though their order is changed.

### 2.2. EO Analysis

#### 2.2.1. Mentheae Tribe

The percentage of identified fraction varied from 84.7% in *S. dorisiana* to being totally identified in both *A. aurantiaca* and Ob-Ct. Appendix A lists the compounds identified by GC-MS in the EOs of the studied species. The prevailed chemical class in the *Agastache* species was OM with an amount ranging from 75.1% in *A*. ‘Blue Boa’ to 88.8% in *A. aurantiaca*, except for *A.* ‘Arcado Pink’, where SH was the main class (58.1%). Pulegone (N° 18) was the constituent with higher percentage in both *A. aurantiaca* and *A*. ‘Blue Boa’ (77.9% and 33.8%, respectively). The latter plant showed a high value of isomenthone too (N° 11; 31.6%). Pulegone was completely absent in *A. mexicana*, where it was substituted by geranyl acetate (N° 34; 24.8%). *A. mexicana* showed other compounds with a good percentage such as citral (neral (N° 19; 12.0%) and geranial (N° 22; 16.6%)), geraniol (N° 21; 10.6%) and citronellol (N° 16; 11.9%). Germacrene D (N° 44; 33.9%) and germacrene D-4-ol (N° 54) were the main compounds in *A.* ‘Arcado Pink’. To the best of our knowledge, no report was present in the literature on the EO of the studied species; however, Haiyan [29] studied the leaf EO of *A. rugosa* and reported that menthan-3-one (48.5) and estragole (20.8%) were from major compounds. The latter (estragole) was also the main individual in the leaf EOs of this same species investigated by Zielińska [30]. This result disagrees with what was found herein, where these two compounds were completely lacking. Otherwise, the *A. mexicana* results agreed with those reported in the leaf EO of *A. astromontana* ‘Pink Pop’ [31] due to the presence of linalool and bornyl acetate. However, they contradicted the results found in the EO of the aerial part of the same species studied by Juárez [32], who underlined the predominance of estragole and methyl eugenol.

*M. didyma*, as the majority of plants in *Agastache* genus, was rich in OMs (88.4%) with linalool (N° 6; 39.9%), thymol (N° 26; 24.7%) and thymol methyl ether (N° 17; 17.7%) as the most abundant compounds. This result confirmed those previously reported in the literature [33] and disagreed with those stated by Gontar [34], where carvacrol (26.2%) and *p*-cymene (11.6%) were of the main compounds in the leaves collected in the flowering period. The leaf juice from this species was reported to treat eczema, burns, and hair loss [35].

Sesquiterpenes dominated in the EO of *N. × faasenii* too, especially the oxygenated one (44.3% vs. 29.6). In detail (*Z*)-α-*trans*-bergamotol (N° 74; 14.1%) and β-caryophyllene (N° 37; 12.4%) were the main individuals in these classes. It is interesting to note the presence of eucalyptol (10.0%) and *cis*, *cis*-nepetalactone (N° 25; 9.1%), two OM compounds of important value. The latter one is a characteristic constituent of this species. A previous study reported that this compound was a major one in the leaf EO of the same species [36,37].

*Salvia* samples showed a heterogeneous behavior because SH (64.5), OM (45.5%) and OS (76.0%) prevailed in *S. discolor*, *S. dorisiana* and *S. microphylla*, respectively. β-caryophyllene (30.9%), perillyl acetate (N° 39; 25.3%) and guaiol (N° 56; 28.9%) were the main constituents in the cited species, respectively. Guaiol (24.6–26.3%) was also reported to be a main compound in *S. microphylla* investigated by Satyal [38] together with α-eudesmol (15.6–19.9%) and (*E*)-caryophyllene (5.5–11.5%). The latter constituent characterized the Algerian species too (17.68%) [39] and all the cited compounds were found herein. The work of Sharopov et al. [40] on *S. discolor* reported also (*E*)-caryophyllene (17.81%) as main constituent completely in agreement with the presented results but in two-fold percentage. The areal part EO of *S. dorisiana* was reported to be characterized by perillyl acetate (21.74%), methyl perillate (19.16%), β-caryophyllene (9.99%), and myrtenyl acetate (N° 28) (4.03%) [41]. All these compounds were present in the leaf EO of *S. dorisiana* studied herein, except for the perillyl acetate.

Looking at the PC2 of the studied EOs, *A.* ‘Blue Boa’ and *A. aurantiaca* were distinguished from the others as in SPME (Figure 3). *M. didyma* was located in the left high quadrant due to its percentage in thymol and thymol methyl ether. In the same quadrant, we found *A. mexicana* and *S. microphylla,* and the latter was almost superposed on the PC1 axis. The left low quadrant encompasses two of Salviinae and Nepetinae member, owed to their good amount of β-caryophyllene.

Two-way HCA (Figure 4) confirmed the PCA results; in fact, the cluster can be divided into two subclusters as was PC2 (Figure 3). The first cluster “I” was homogenous because both *A. aurantiaca* and *A.* ‘Blue Boa’ were members of the same subtribe. The second one “II”, where we can find the species positioned in the negative loading of PC2, was divisible into two subclusters: “IIa” which combined *M. didyma* with *A. mexicana* and *S. microphylla,* while “IIb” included the remaining ones.

#### 2.2.2. Ocimeae Tribe

Regarding the EOs from *Ocimum* varieties, *O. basilicum* ‘Cinnamon’ (Ob-Cn) as well as *O. × citriodorum* (Ob-Ct) were noted for the good amount of SHs which represented almost 50% of the identified fraction even though the main constituent in Ob-Cn was τ-cadinol (N° 62; 22.4%), an OS compound. Moreover, the latter was rich in germacrene D (14.6%) and α-bulnesene (N° 48; 10.0%), instead of germacrene B (N° 52; 22.1%) and β-bisabolene (N° 47; 18.7%), which were the main compounds in Ob-Cn. It is interesting to note the presence of phenylpropanoids in these two species, mainly represented by eugenol (9.5% in Ob-Cn vs. 29.9% in Ob-Ct). Geranial (44.2%), neral (33.8%), and tetracosane (N° 83) (11.5%) were dominant in Ob-BS. According to Lal [42], an EO rich in citral is in high demand today, and so a short duration *Ocimum africanum* (= *O. × citrodorum*, hairy basil) variety rich in citral (>65% *w/v*) was developed at CSIR-CIMAP (Central Institute of Medicinal and Aromatic Plants, Lucknow (India)). The leaves of this hairy basil are widely consumed in Thailand and eaten as a side-dish, or used as an ingredient in curry to increase lactation and used in traditional medicine as a carminative, to enhance digestion, and as remedy for gastrointestinal disturbance [43]. The composition of leaf EO from the Jordan sweet basil (*O. basilicum* ‘Cinnamon’) was studied by Aburjai et al. [44] who reported linalool (36.26%), eugenol (14.2%), eucalyptol (11.36%), and *trans-*α-bergamotene (9.00%) as main constituents, while Majdi and co-workers [45] reported a high amount of linalool (26.5.5%) and (*E*)-methyl-cinnamate (24.7%). All these compounds are present in our study but in a lesser amount, except for (*E*)-methyl-cinnamate. No reports are present in the literature on the composition of Ob-BS leaf EO; thus, this is the first report thereof.

The one-way PERMANOVA (Table 4A) statistical test confirmed a significant difference between subtribes, even though their pairwise results were unable to distinguish between them (Table 5A1). On the contrary, the same test performed on the tribe’s effect underlined a significant difference between the Ocimeae and Mentheae tribes (Table 4B and Table 5B1) and half of this dissimilarity was due to ten compounds of which geranial eugenol and neral stood out (Table 6).

Many of the above-cited compounds are sought after for their biological value and valorize the leaves, a waste product of the supply chain of EFs. Among them, we can find β-caryophyllene, a compound approved by the Food and Drug Administration (FDA) and European Food Safety Authority (EFSA) for use as flavor enhancer [46] and in cosmetics too [47]. It had 5- and 25-fold more leaf EOs of *S. discolor* and *N. × faasenii* than in the flowers of the same species, respectively [48]. On the other hand, germacrene D, a compound detected only in the leaf EO of each of *A.* ‘Arcado Pink’, *N. × faasenii*, and *M. didyma* (totally absent in flowers) with 8-fold more leaf EO of *S. discolor* [48], is involved in plant-insect interaction acting as a pheromone on receptor neurons, also used for its deterrent and insecticidal activities against different parasites, such as mosquitos, aphids, and ticks, and has antimicrobial properties [49,50].

The microbial drug resistance problem boosted the research on natural compounds, and among these we can cite pulegone, a constituent known for its proven antibacterial and anti-inflammatory activities that, although present in a higher amount in the flower EOs, its percentages exceed 30% and 70% in the leaf EOs of each *A.* ‘Blue Boa’ and *A. aurantiaca*, respectively [51,52]. Linalool, a monoterpene unsaturated alcohol, instead, is one of the major floral scents in nature [53], widely used in the perfumery, cosmetics, and food industries [54]. The leaf EO of *M. didyma* can be a good source, since its value was quadrupled in comparison with flowers and constitutes almost 40% of the identified fraction. Furthermore, citral, with its two isomers (neral and geranial), generally present in the lemongrass oil at a concentration of 65–85%, is also used in the perfumery industry due to its sweet citrus smell [55]. Found exclusively in the EO from Ob-BS leaves, it showed a similar amount. However, its presence also in exclusivity and with a very high percentage in *A. mexicana* and OB-Ct is not to be underestimated. Leaves of these plants, besides having decorative purpose and the food use of their flowers, can be introduced to the cosmetics and perfumery industries. Furthermore, eugenol can also be used in both the food and perfumery industries; in fact, it is used as a flavor in the food industry, in addition to its role in the enhancement of the skin penetration of different drugs or to treat skin infections [56,57]. Even though the amount of eugenol obtained by Ob-Ct is less than the amount reported in *Syzygium aromaticum* (L.) Merr. & L.M.Perry, this does not deny the possible use of the leaves of this species in different industries. The known inhibitory effects of τ-cadinol on intestinal hypersecretion and ileum contractions as well as its calcium antagonist properties [58] could make Ob-Cn leaves a good candidate to treat digestive disorders.

## 3. Materials and Methods

### 3.1. Plant Material and Cultivation

The plants were provided and cultivated as reported in our previous work [41,48,59]. Briefly, *Monarda didyma* ‘Fireball’, *Nepeta × faasenii* ‘Six Hills Giant’ were bought at L’Erbario della Gorra (Str. Gianardo, 11 Casalborgone, TO, Italy), while *Salvia discolor* and *S. microphylla* ‘Hot Lips’ were part of the CREA plant collection in Sanremo (CREA, Research Centre for Vegetable and Ornamental Crops, Sanremo, IM, Italy; GPS: 43.816887, 7.758900). S. *dorisiana* was instead derived from cutting then cultivated in pot. *Ocimum* varieties and hybrids (*O. basilicum* ‘Blue Spice’(Ob-BS), *O. basilicum* ‘Cinnamon’ (Ob-Cn) and *O. × citriodorum* (Ob-Ct)) were grown from seeds, provided by the Conservatoire National des Plantes à Parfum, Medicinales et Aromatiques (Milly-la-Forêt, France), as well as *A.* ‘Blue Boa’and *A. aurantiaca* ‘Sunset Yellow’. *A. mexicana* ‘Sangria’ and *A.* ‘Arcado Pink’) seed, instead, were provided by the Chambre d’Agriculture des Alpes-Maritimes (CREAM, Nice, France) (Figure 5). All of these species were grown in pots at CREA with slow-release fertilizer, and weekly irrigated with a nutrient solution. The pots were placed in greenhouses with an anti-insect net, and the plants were cultivated organically.

### 3.2. Phytochemical Survey

HS-SPME analysis follows the method previously reported [48] with minor modifications. 1 g of leaves were placed separately into 50 mL sealed vials. The samples were extracted at room temperature (around 24 °C) for 30 with a preconditioned solid phase microextraction device (SPME, Supelco, Bellefonte, PA, USA) coated with polydimethylsiloxane (PDMS, 100 μm coating thickness, St. Louis, MO, USA). Fresh leaves were subjected to EO extraction which was performed, as recommended by the European Pharmacopoeia, by hydrodistillation for 2 h in a Clevenger apparatus. The EOs were collected with *n*-hexane (HPLC grade) and immediately injected into the GC-MS, due to the low amount the EO produced as an upshot of the low quantities of the starting plant material at our disposal (only 5–10 g). Both Volatile Organic Compounds (VOCs) and EO composition were performed with a GC-MS instrument and the equipment description and conditions of use, together with the identification method of compounds, were already reported in previous studies [41,48,59].

### 3.3. Statistical Analysis

A variance-covariance matrix of 1677 (in VOC) and 1579 (in EO) variants in the dataset was used for the principal component analysis (PCA) and two-way hierarchical cluster analysis (HCA). These analyses concerned only the Mentheae tribe. Briefly, these matrixes were used for the measurement of eigenvalues and eigenvectors in PCA analysis, where the plot was performed selecting the two highest principal components (PCs). Two-way HCA was performed using Ward’s method with squared Euclidian distances. The statistically significant differences induced by tribe and subtribe on VOC and EO composition were assessed by PERMANOVAs (Permutational Multivariate Analysis of Variance) with the Bray-Curtis dissimilarity index, which is the method to measure the dissimilarities between different groups and it is based on a distribution-free analysis of variance. The percentage contribution of each compound to the observed dissimilarity was assessed through similarity percentage analysis (SIMPER, Euclidean distance). All statistical analyses were performed with Past 4 software.

## 4. Conclusions

The edible flower supply chain increased in recent years due to the high demand for new food ingredients or additives. Like any other supply chain, the waste of this one is related to leaves in the first row. This work reported the volatile profile of the leaves from some Lamiaceae species investigated during the Interreg ALCOTRA “ANTEA” project (N° 1139). The spontaneous emission of the studied species is reported here for the first time except for *A*. ‘Blue Boa’and *O. × citriodorum*. The Nepetinae subtribe was characterized by a high percentage in OM except for *A.* ‘Arcado Pink’ and *N. × faasenii,* where the main class was SH. This one was also the predominant class in each of the Ociminae, Menthinae and Salviinae subtribes with the exception of *S. dorisiana,* which was characterized by OM. Herein is the first report of *Agastache* EO composition as well as *O. basilicum* ‘Blue Spice’. The Nepetinae EO showed a similar behavior to their VOC with a switch of SH in OS in *N. × faasenii*. In contrast, the other species underlined a change in their profile in comparison with what was found in VOC. Consequently, our results offer not only information on the chemical composition of these plant organs but could contribute to the possible use of this waste product, and to introduce these leaves or their derivatives (EOs) in other new industries due to their richness of active compounds.

## Figures and Tables

**Figure 1 molecules-27-02172-f001:**
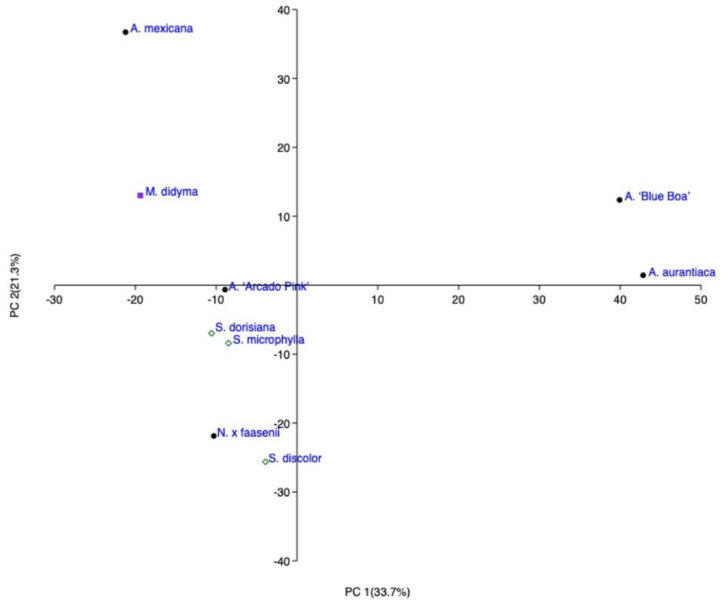
Principal Cluster Analysis (PCA) of the spontaneous emission of the plants belonging to the Mentheae tribe.

**Figure 2 molecules-27-02172-f002:**
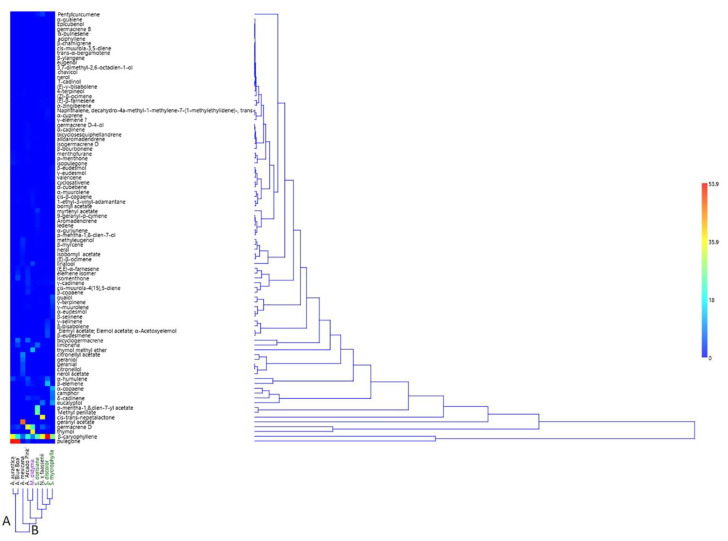
Two-way Hierarchical Cluster Analysis (HCA) of the spontaneous emission of the plants belonging to the Mentheae tribe.

**Figure 3 molecules-27-02172-f003:**
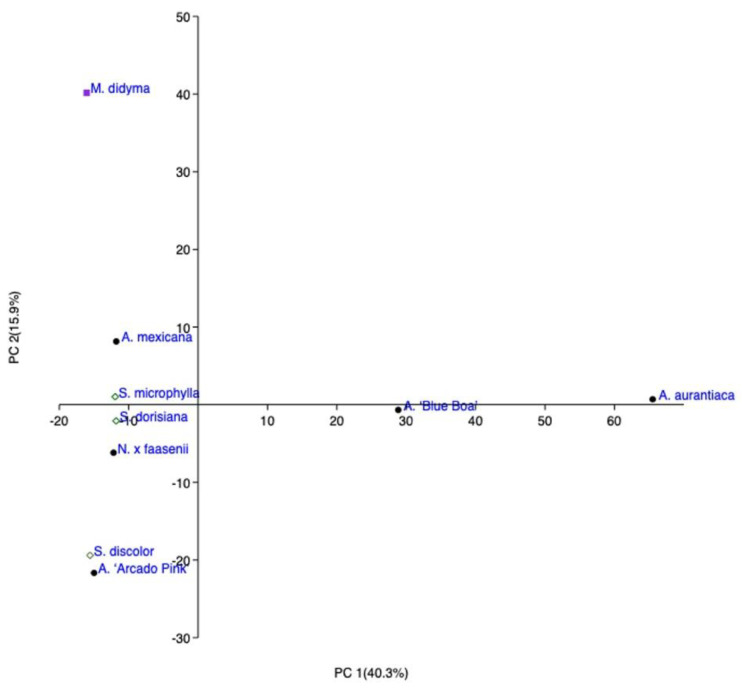
Principal Cluster Analysis (PCA) of the EOs obtained from all the studied plants belonging to the Mentheae tribe.

**Figure 4 molecules-27-02172-f004:**
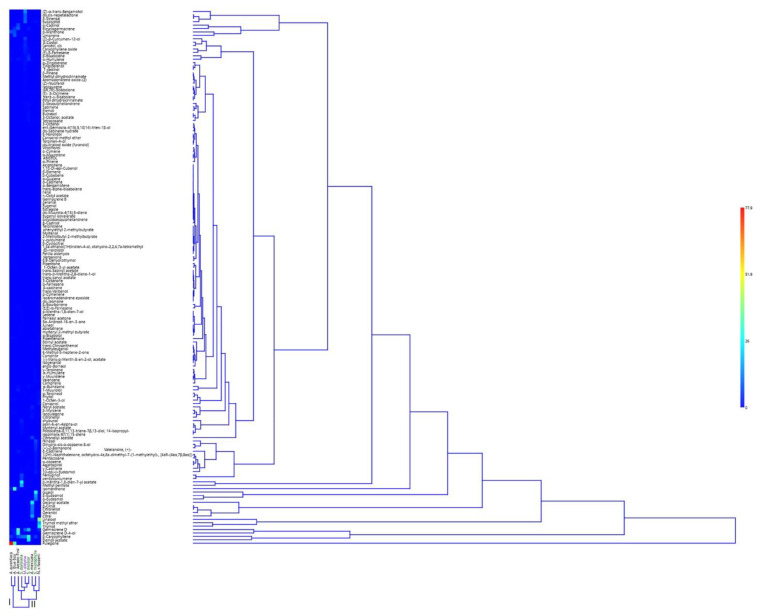
Two-way Hierarchical cluster Analysis (HCA) of the EOs obtained from all the studied plants belonging to the Mentheae tribe.

**Figure 5 molecules-27-02172-f005:**
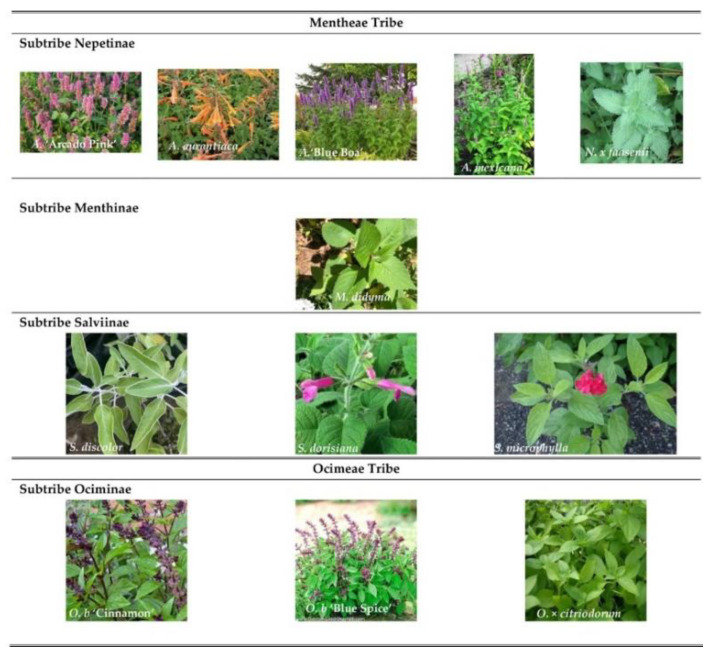
Selected flowers belonging to two tribe of Lamiaceae family.

**Table 1 molecules-27-02172-t001:** Effects of Subtribes (**A**) and Tribes (**B**) on the Volatile Organic Compounds (VOCs) according one-way PERMANOVA analysis.

A		B	
Permutation N:	9999	Permutation N:	9999
Total sum of squares:	4.629	Total sum of squares:	6.007
Within-group sum of squares:	3.282	Within-group sum of squares:	4.727
F:	3.079	F:	4.332
*p* (same):	**0.0018**	*p* (same):	**0.0002**

**Bold form:** *p*-value is significant.

**Table 2 molecules-27-02172-t002:** Pairewise of one-way PERMANOVA test (Bonferroni-corrected *p*- value) of Subtribe (**A1**) and (**B1**) on VOCs.

A1				B1		
	Subtribe Nepetinae	Subtribe Menthinae	Subtribe Salviinae		Mentheae Tribe	Ocimeae Tribe
**Subtribe Nepetinae**		0.0975	**0.0327**	**Mentheae Tribe**		0.0001
**Subtribe Menthinae**	0.0975		0.1119	**Ocimeae Tribe**	**0.0001**	
**Subtribe Salviinae**	**0.0327**	0.1119				

**Bold form:** *p*-value is significant.

**Table 3 molecules-27-02172-t003:** SIMPER analyses test on subtribes (**A2**) and tribes (**B2**) dissimilarity on VOCs.

A2							B2						
	Compounds	Av. dissim	Contrib. %	Cumulative %	Mean Nep.Str	Mean Salv.Str		Compounds	Av. dissim	Contrib. %	Cumulative %	Mean Men.Tr	Mean Oci.Tr
**1**	Pulegone	10.3	14.1	14.1	20.3	0.0	**1**	β-Caryophyllene	11.9	13.2	13.2	26.7	3.4
**2**	β-Caryophyllene	8.2	11.2	25.3	25.1	35.2	**2**	Eugenol	11.2	12.4	25.6	0.0	22.1
**3**	Geranyl acetate	4.7	6.5	31.8	9.3	0.0	**3**	Pulegone	5.7	6.3	31.9	11.3	0.0
**4**	Germacrene D	4.3	5.8	37.6	8.9	0.8	**5**	β-Bisabolene	4.5	5.0	42.5	0.5	8.8
**5**	Methyl perillate	4.2	5.8	43.4	0.0	8.4	**6**	Germacrene B	4.2	4.7	47.2	0.0	8.3
**7**	*cis-trans*-Nepetalactone	3.8	5.2	54.1	7.4	0.0	**7**	Geranial	4.1	4.5	51.7	0.6	7.9
**8**	β-Elemene	2.4	3.3	57.4	0.7	4.5	**8**	Germacrene D	4.1	4.5	56.2	7.8	5.3
**9**	Eucalyptol	2.1	2.9	60.3	1.5	3.7	**9**	Geranyl acetate	2.6	2.9	59.1	5.2	0.0
**10**	α-Copaene	2.1	2.8	63.1	0.1	4.0	**10**	β-Elemene	2.4	2.7	61.8	2.1	4.1
**11**	δ-Cadinene	1.6	2.2	65.3	0.8	3.3	**11**	γ-Cadinene	2.3	2.5	64.3	0.7	4.3
**12**	Bicyclogermacrene	1.5	2.1	67.4	3.0	0.0	**12**	Thymol	2.1	2.3	66.6	4.1	0.0
**13**	Camphor	1.4	1.9	69.3	0.0	2.8	**13**	*cis-trans*-Nepetalactone	2.1	2.3	68.9	4.1	0.0
**14**	Limonene	1.4	1.9	71.2	1.3	2.3	**14**	Neral	1.5	1.6	70.5	0.2	2.9
**15**	α-Humulene	1.3	1.8	73.0	2.6	2.9	**15**	Methyl perillate	1.4	1.6	72.1	2.8	0.0
**16**	Guaiol	1.0	1.3	74.3	0.0	1.9	**16**	T-Cadinol	1.4	1.5	73.6	0.0	2.7
**17**	Citronellyl acetate	0.9	1.3	75.6	1.9	0.0	**17**	*p*-Mentha-1,8,dien-7-yl acetate	1.3	1.5	75.1	2.6	0.0
							**18**	δ-Guaiene	1.3	1.4	76.5	0.0	2.6
							**19**	*trans*-α-Bergamotene	1.2	1.4	77.9	0.0	2.5
							**20**	eucalyptol	1.1	1.2	79.1	2.1	0.1
							**21**	α-humulene	1.0	1.1	80.2	2.4	1.9

Nep.Str: Nepetinae Subtribe; Salv.Str: Salviinae Subtribe; Men.Tr: Mentheae Tribe; Oci.Tr: Ocimeae Tribe.

**Table 4 molecules-27-02172-t004:** Effects of Subtribes (**A**) and Tribes (**B**) on the essential oils (EOs) according one-way PERMANOVA analysis.

A		B	
Permutation N:	9999	Permutation N:	9999
Total sum of squares:	6.559	Total sum of squares:	6.912
Within-group sum of squares:	4.75	Within-group sum of squares:	5.994
F:	2.857	F:	2.451
*p* (same):	**0.0012**	*p* (same):	**0.0014**

**Bold form:** *p*-value is significant.

**Table 5 molecules-27-02172-t005:** Pairwise of one-way PERMANOVA test on the EOs of Subtribes (**A1**) and Tribes (**B1**).

A1				B1		
	Subtribe Nepetinae	Subtribe Menthinae	Subtribe Salviinae	Mentheae Tribe	Ocimeae Tribe	Mentheae Tribe
**Subtribe Nepetinae**		0.0885	0.0942	**Mentheae**		**0.0024**
**Subtribe Menthinae**	0.0885		0.1011	**Ocimeae**	**0.0024**	
**Subtribe Salviinae**	0.0942	0.1011				

Bonferroni-corrected *p*-value.

**Table 6 molecules-27-02172-t006:** SIMPER analyses test on subtribe (**A2**) and tribe (**B2**) dissimilarity performed in essential oils composition.

	Compounds	Av. dissim	Contrib. %	Cumulative %	Mean Mentheae Tribe	Mean Ocimeae Tribe
**1**	Geranial	7.5	8.1	8.1	0.0	14.7
**2**	Eugenol	6.8	7.3	15.4	0.0	13.2
**3**	Neral	6.5	7.0	22.4	0.0	12.8
**4**	β-Caryophyllene	4.9	5.2	27.6	10.3	1.5
**5**	Linalool	4.3	4.6	32.2	7.3	2.8
**6**	T-Cadinol	3.9	4.2	36.4	0.2	7.5
**7**	Germacrene B	3.8	4.1	40.5	0.0	7.4
**8**	Germacrene D	3.7	4.0	44.5	5.4	6.4
**9**	β-Bisabolene	3.4	3.6	48.1	0.8	6.2
**10**	Guaiol	2.4	2.6	50.7	4.8	0.0
**11**	*p*-Mentha-1,8-dien-7-yl acetate	2.3	2.5	53.2	4.2	0.0
**12**	Geranyl acetate	2.1	2.3	55.5	4.2	0.0
**13**	Thymol	2.1	2.2	57.7	4.1	0.0
**14**	Elemol acetate	2.1	2.2	59.9	4.1	0.0
**15**	Tetracosane	2.0	2.2	62.1	0.4	3.9
**16**	α-Bulnesene	1.7	1.9	64.0	0.0	3.3
**17**	Methyl perillate	1.6	1.7	65.7	2.9	0.0
**18**	Thymol methyl ether	1.5	1.6	67.3	3.0	0.0
**19**	β-Eudesmol	1.4	1.5	68.8	2.6	0.3
**20**	Citral	1.4	1.5	70.3	2.8	0.0
**21**	Estragole	1.3	1.4	71.8	0.0	2.6
**22**	γ-Cadinene	1.3	1.4	73.2	0.4	2.4
**23**	(*Z*)-α-*trans*-Bergamotol	1.2	1.3	74.5	2.3	0.0
**24**	α-Humulene	1.1	1.2	75.7	1.2	2.3
**25**	Eucalyptol	1.1	1.2	76.8	2.0	0.5
**26**	β-Citral	1.0	1.1	77.9	2.0	0.0
**27**	Citronellol	1.0	1.1	79.0	2.0	0.0
**28**	α-Eudesmol	1.0	1.1	80.1	2.0	0.0

## Data Availability

The data presented in this study are available in Appendix A.

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
