# Peer review of "Valorization of a Waste Product of Edible Flowers: Volatile Characterization of Leaves"

_molecules, 2022, doi:10.3390/molecules27072172_

Round 1
Reviewer 1 Report
The researchers conducted a detailed analysis of the differences in VP and EO of leaves of 12 Lamiaceae plants. However, we believe that there are still some problems in this paper. The following are the questions in this manuscript:
- From the preface to the content of line 152, the format is not uniform with the following.
- Figures 1 and 3 are too blurry, and the resolution should be adjusted.The position of the data labels should not obscure the data axis.
- Tables 1, 2, 4 and 5 are hard to understand.
- Is it clear that citral (neral (N°17)+geranial (N°21)) is stated in the fourth line?They should belong to the oxygenated monoterpenes.
- We believe that the literature should also be consulted to compare the differences in the main chemical constituents of the flowers and leaves of the 12 species of Lamiaceae.
- The functions of the main chemical components should also be briefly introduced so that the reader can better understand them.
- There is a problem with the fourth reference.
Author Response
COMMENTI REVIEWER 1
Open Review
(x) I would not like to sign my review report
( ) I would like to sign my review report
English language and style
( ) Extensive editing of English language and style required
(x) Moderate English changes required
( ) English language and style are fine/minor spell check required
( ) I don't feel qualified to judge about the English language and style
|
Yes |
Can be improved |
Must be improved |
Not applicable |
|
|
Does the introduction provide sufficient background and include all relevant references? |
(x) |
( ) |
( ) |
( ) |
|
Is the research design appropriate? |
(x) |
( ) |
( ) |
( ) |
|
Are the methods adequately described? |
( ) |
(x) |
( ) |
( ) |
|
Are the results clearly presented? |
( ) |
(x) |
( ) |
( ) |
|
Are the conclusions supported by the results? |
(x) |
( ) |
( ) |
( ) |
The researchers conducted a detailed analysis of the differences in VP and EO of leaves of 12 Lamiaceae plants. However, we believe that there are still some problems in this paper. The following are the questions in this manuscript:
1. From the preface to the content of line 152, the format is not uniform with the following.
Answer: we correct the form of the plant name. Now is in italic form.
2. Figures 1 and 3 are too blurry, and the resolution should be adjusted. The position of the data labels should not obscure the data axis.
Answer: we tried to increase the resolution as recommended by Molecule’s guideline
3. Tables 1, 2, 4 and 5 are hard to understand.
Answer: tables 1,2,4 and 5 contain the results of a one-way PERMANOVA test. It is a statistical test that shows the presence or absence of a significant difference in the composition of VOC (table 1) and EO (table 4). p-value is less than 0.05 gradient in the case of table 1 means that there is a significant difference between subtribes (table 1A, p-value was: 0.0018) and tribes (table 1B: p-value was 0.0002). However, table 2 and 5 contain the results of the pairewise of PERMANOVA test which underlined between which subtribe or tribe this difference is noted. We add a sentence in the material and methods section 3.3 statistical analysis
4. Is it clear that citral (neral (N°17)+geranial (N°21)) is stated in the fourth line? They should belong to the oxygenated monoterpenes.
Answer: We agree that the sentence is formulated in a wrong way since, of course, citral belongs to oxygenated monoterpenes. Please see the revised sentence
5. We believe that the literature should also be consulted to compare the differences in the main chemical constituents of the flowers and leaves of the 12 species of Lamiaceae.
Answer: Thank you for your recommendation. To the best knowledge of the authors no spices were reported in the literature for their spontaneous emissions on the studied species except for Ocimum × citriodorum. Only few works were reported for other species of the same genus and was included in the manuscript. About the EOs, only five out of 12 species were reported in the literature, and we cited those more recent ones.
6. The functions of the main chemical components should also be briefly introduced so that the reader can better understand them.
Answer: The functions of the reported compounds and their possible use in several industries were discussed at the end of the results and discussion section.
7. There is a problem with the fourth reference
Answer: Done

Reviewer 2 Report
In Table S1, 'calss' should be corrected to 'class'.
Author Response
COMMENTI REVIEWER 2
Open Review
(x) I would not like to sign my review report
( ) I would like to sign my review report
English language and style
( ) Extensive editing of English language and style required
( ) Moderate English changes required
( ) English language and style are fine/minor spell check required
(x) I don't feel qualified to judge about the English language and style
|
Yes |
Can be improved |
Must be improved |
Not applicable |
|
|
Does the introduction provide sufficient background and include all relevant references? |
(x) |
( ) |
( ) |
( ) |
|
Is the research design appropriate? |
(x) |
( ) |
( ) |
( ) |
|
Are the methods adequately described? |
(x) |
( ) |
( ) |
( ) |
|
Are the results clearly presented? |
(x) |
( ) |
( ) |
( ) |
|
Are the conclusions supported by the results? |
(x) |
( ) |
( ) |
( ) |
Comments and Suggestions for Authors
In Table S1, 'calss' should be corrected to 'class'.
Answer: Done

Reviewer 3 Report
Reviewers' comments:
Manuscript Number: molecules-1629637
Title: Valorisation of a waste product of edible flowers: volatile characterization of leaves
The MS deals with the valorisation of a waste product of edible flowers: volatile characterization of leaves. The work is interesting since it is reporting the utilization of natural waste products of edible flowers. I feel that the paper in the present form lacks the quality needed for this journal, and key experiments /details are missing. However, I have several significant concerns to be addressed, my comments and suggestions are listed below:
English editing is needed for this manuscript. Several statements should be re-written, Authors have to carefully check the English grammar throughout the manuscript.
Comments
- Materials and experimental section (for eg: section 3.2, HS-SPME analysis) should be improved by providing the information in detail for better understanding.
- Conclusion: author should also highlight the outcome/results of this study, concisely
- Please improve the quality of Fig. 1, Fig 2, Fig 3, Fig 4, and provide the new figure with good quality.
- Line 96-100 and 145-150: The author should explain in detail the HAC analysis (fig 2 and 4). It's not clear, A and B cluster. Please mark and highlight in the figure as well.
- Line 57: Ocimum basilicum, italicize the scientific name
- Line 62: A. foeniculum, italicize the scientific names throughout the manuscript.
- Line 70: form change to from.
- Line 78, 79, 82, 113, etc., italicize
- Line 94: mentioned as characterised, but in line 24 mentioned as characterized, please check the grammar throughout the manuscript.
- Line 6-12: It is noteworthy evidenced the high amount of geranial (N°21), an oxygenated monoterpene compound, in Ob-Ct (23.7%). Many were the papers on the spontaneous emission of Ocimum basilicum (Ob) aerial parts [21][22][23][24][25][26]. The high presence of eugenol (41.9%) was also observed by Tirillini and Maggi [27]. However, Al-Kateb [28] investigating the lemon basil (O. × citriodorum), showed that the citral amount, the major contributor for the lemony flavour characteristic of lemon basil infusions, was considerably higher in full-flowering stage (65%). Please revise this paragraph.
- In result and discussion: Author should improve the result and discussion part by explain the results in detail for better understanding.
- Author should include the sub-section for different category in the result and discussion part and explain it with more clarity.
- Include more relevant references and discuss, elaborately.
For eg: Perumal et al., 2016; Antifungal activity of five different essential oils in vapour phase for the control of Colletotrichum gloeosporioides and Lasiodiplodia theobromae in vitro and on mango
Perumal et al., 2017; Effects of Essential Oil Vapour Treatment on the Postharvest Disease Control and Different Defence Responses in Two Mango (Mangifera indica L.) Cultivars
Perumal, et al., 2021. Preparation and characterization of a novel green tea essential oil nanoemulsion and its antifungal mechanism of action against Magnaporthae oryzae. Ultrasonics Sonochemistry, 76, 105649.
Perumal et al., 2022; Application of essential oils in packaging films for the preservation of fruits and vegetables: A review.
- Line 52: later?
- Line 81-83: Previous study by our team on areal part of S. dorisiana EO pointed out that this species was characterised by perillyl acetate (21.74%) and methyl perillate (19.16%), together with β-caryophyllene (9.99%) and myrtenyl acetate (N°28) (4.03%) [41]. Revise the sentence.
- Line 156: outdo?
- Table 7: Name of the plants are missing
Author Response
COMMENTI REVIEWER 3
Open Review
( ) I would not like to sign my review report
(x) I would like to sign my review report
English language and style
( ) Extensive editing of English language and style required
(x) Moderate English changes required
( ) English language and style are fine/minor spell check required
( ) I don't feel qualified to judge about the English language and style
|
Yes |
Can be improved |
Must be improved |
Not applicable |
|
|
Does the introduction provide sufficient background and include all relevant references? |
(x) |
( ) |
( ) |
( ) |
|
Is the research design appropriate? |
(x) |
( ) |
( ) |
( ) |
|
Are the methods adequately described? |
() |
( ) |
(x) |
( ) |
|
Are the results clearly presented? |
() |
( ) |
(x) |
( ) |
|
Are the conclusions supported by the results? |
() |
(x) |
( ) |
( ) |
Comments and Suggestions for Authors
Reviewers' comments:
Manuscript Number: molecules-1629637
Title: Valorisation of a waste product of edible flowers: volatile characterization of leaves
The MS deals with the valorisation of a waste product of edible flowers: volatile characterization of leaves. The work is interesting since it is reporting the utilization of natural waste products of edible flowers. I feel that the paper in the present form lacks the quality needed for this journal, and key experiments /details are missing. However, I have several significant concerns to be addressed, my comments and suggestions are listed below:
English editing is needed for this manuscript. Several statements should be re-written, Authors have to carefully check the English grammar throughout the manuscript.
Comments
- Materials and experimental section (for eg: section 3.2, HS-SPME analysis) should be improved by providing the information in detail for better understanding.
Answer: Thank you for you observation. The method was reported in our previous study (reference 48) and to avoid paganism, we just cited the slight modification done on that method
- Conclusion: author should also highlight the outcome/results of this study, concisely
Answer: A recap regarding the result found in this work was added in the conclusion
- Please improve the quality of Fig. 1, Fig 2, Fig 3, Fig 4, and provide the new figure with good quality.
Answer: the resolution of figures was improved according to the journal guideline
- Line 96-100 and 145-150: The author should explain in detail the HAC analysis (fig 2 and 4). It's not clear, A and B cluster. Please mark and highlight in the figure as well.
Answer: Thank you for your comment. We added sentences in material and methods section 3.3 statistical analysis to explain more each statistical analysis test. The names of each cluster were also added in the respective figures
- Line 57: Ocimum basilicum, italicize the scientific name
Answer: Done
- Line 62: A. foeniculum, italicize the scientific names throughout the manuscript.
Answer: Done
- Line 70: form change to from.
Answer: Done
- Line 78, 79, 82, 113, etc., italicize
Answer: Done
- Line 94: mentioned as characterised, but in line 24 mentioned as characterized, please check the grammar throughout the manuscript.
Answer: Thanks for your comment. It depends on whether American or British English was used. In any case we have written it in the same way throughout the manuscript.
- Line 6-12: It is noteworthy evidenced the high amount of geranial (N°21), an oxygenated monoterpene compound, in Ob-Ct (23.7%). Many were the papers on the spontaneous emission of Ocimum basilicum (Ob) aerial parts [21][22][23][24][25][26]. The high presence of eugenol (41.9%) was also observed by Tirillini and Maggi [27]. However, Al-Kateb [28] investigating the lemon basil (O. × citriodorum), showed that the citral amount, the major contributor for the lemony flavour characteristic of lemon basil infusions, was considerably higher in full-flowering stage (65%). Please revise this paragraph.
Answer: We reformulate the paragraph and we hope that it is now more clear
- In result and discussion: Author should improve the result and discussion part by explain the results in detail for better understanding.
Answer: Sorry but we cannot understand what you mean by “explain the results in detail”, please be clear to help us to improve the manuscript.
- Author should include the sub-section for different category in the result and discussion part and explain it with more clarity.
Answer: Subsections were added
- Include more relevant references and discuss, elaborately.
For eg: Perumal et al., 2016; Antifungal activity of five different essential oils in vapour phase for the control of Colletotrichum gloeosporioides and Lasiodiplodia theobromae in vitro and on mango
Perumal et al., 2017; Effects of Essential Oil Vapour Treatment on the Postharvest Disease Control and Different Defence Responses in Two Mango (Mangifera indica L.) Cultivars
Perumal, et al., 2021. Preparation and characterization of a novel green tea essential oil nanoemulsion and its antifungal mechanism of action against Magnaporthae oryzae. Ultrasonics Sonochemistry, 76, 105649.
Perumal et al., 2022; Application of essential oils in packaging films for the preservation of fruits and vegetables: A review.
Answer: Thank you for your suggestion, but we are disagreed with the referee to cite the suggested references because they are not inherent with our manuscript topic. Could you please suggest us one more inherent with the work, we will be a pleasure to cite it and improve our discussion.
- Line 52: later?
Answer: Sorry, it was a typing error: the word was replaced by “latter”
- Line 81-83: Previous study by our team on areal part of S. dorisiana EO pointed out that this species was characterised by perillyl acetate (21.74%) and methyl perillate (19.16%), together with β-caryophyllene (9.99%) and myrtenyl acetate (N°28) (4.03%) [41]. Revise the sentence.
Answer: The sentence was revised and rewritten
- Line 156: outdo?
Answer: This word was changed by with “exceed”
- Table 7: Name of the plants are missing
Answer: Sorry and thank you for your observation. The name of the plant was added

Round 2
Reviewer 3 Report
Figure 2 and Figure 4 are still of poor quality, the author should improve the figures.
Novelty and the potential application should be emphasized in the introduction part of this manuscript.
Author Response
Open Review
( ) I would not like to sign my review report
(x) I would like to sign my review report
English language and style
( ) Extensive editing of English language and style required
( ) Moderate English changes required
(x) English language and style are fine/minor spell check required
( ) I don't feel qualified to judge about the English language and style
|
Yes |
Can be improved |
Must be improved |
Not applicable |
|
|
Does the introduction provide sufficient background and include all relevant references? |
( ) |
(x) |
( ) |
( ) |
|
Is the research design appropriate? |
(x) |
( ) |
( ) |
( ) |
|
Are the methods adequately described? |
( ) |
(x) |
( ) |
( ) |
|
Are the results clearly presented? |
( ) |
(x) |
( ) |
( ) |
|
Are the conclusions supported by the results? |
(x) |
( ) |
( ) |
( ) |
Comments and Suggestions for Authors
Figure 2 and Figure 4 are still of poor quality, the author should improve the figures.
Answer: Dear Reviewer, we tried our best to increase the resolution of these figures. Unfortunately, they were generated by the Past4 Software, and it didn’t give the possibility to change the resolution. We tried to do it using other software and now we hope is clearer and readable.
Novelty and the potential application should be emphasized in the introduction part of this manuscript.
Answer: We add a small paragraph (highlighted in green) in the introduction where we improved and emphasized the potential use of these leaves.
